# First dose ChAdOx1 and BNT162b2 COVID-19 vaccinations and cerebral venous sinus thrombosis: A pooled self-controlled case series study of 11.6 million individuals in England, Scotland, and Wales

Steven Kerr[1]*, Mark Joy[2], Fatemeh Torabi[3], Stuart Bedston[3], Ashley Akbari[3], Utkarsh Agrawal[4], Jillian Beggs[5], Declan Bradley[6,7], Antony Chuter[5], Annemarie B. Docherty[1], David Ford[3], Richard Hobbs[2], Srinivasa Vittal Katikireddi[8], Emily Lowthian[3], Simon de Lusignan[2], Ronan Lyons[2], James Marple[9], Colin McCowan[4], Dylan McGagh[2], Jim McMenamin[10], Emily Moore[10], Josephine-L. K. Murray[10], Rhiannon K. Owen[2], Jiafeng Pan[11], Lewis Ritchie[12], Syed Ahmar Shah[1], Ting Shi[1], Sarah Stock[1], Ruby S. M. Tsang[2], Eleftheria Vasileiou[1], Mark Woolhouse[1], Colin R. Simpson[1,13], Chris Robertson[10,11], Aziz Sheikh[1]

**1** Usher Institute, The University of Edinburgh, Edinburgh, United Kingdom, **2** Nuffield Department of Primary Care Health Sciences, University of Oxford, Oxford, United Kingdom, **3** Population Data Science, Swansea University Medical School, Swansea, United Kingdom, **4** School of Medicine, University of St. Andrews, St Andrews, United Kingdom, **5** BREATHE–The Health Data Research Hub for Respiratory Health, University of Edinburgh, Edinburgh, United Kingdom, **6** Queen's University Belfast, Belfast, United Kingdom, **7** Public Health Agency, Belfast, United Kingdom, **8** MRC/CSO Social & Public Health Sciences Unit, Glasgow, United Kingdom, **9** Royal Infirmary of Edinburgh, NHS Lothian and Anaesthesia, Critical Care and Pain Medicine, The University of Edinburgh, Edinburgh, United Kingdom, **10** Public Health Scotland, Glasgow, United Kingdom, **11** Department of Mathematics and Statistics, University of Strathclyde, Glasgow, United Kingdom, **12** Academic Primary Care, University of Aberdeen School of Medicine and Dentistry, Aberdeen, United Kingdom, **13** School of Health, Wellington Faculty of Health, Victoria University of Wellington, New Zealand

* steven.kerr@ed.ac.uk

**Data Availability Statement:** The data used in this study cannot be shared publicly because they are

## Abstract

### Background

Several countries restricted the administration of ChAdOx1 to older age groups in 2021 over safety concerns following case reports and observed versus expected analyses suggesting a possible association with cerebral venous sinus thrombosis (CVST). Large datasets are required to precisely estimate the association between Coronavirus Disease 2019 (COVID-19) vaccination and CVST due to the extreme rarity of this event. We aimed to accomplish this by combining national data from England, Scotland, and Wales.

### Methods and findings

We created data platforms consisting of linked primary care, secondary care, mortality, and virological testing data in each of England, Scotland, and Wales, with a combined cohort of 11,637,157 people and 6,808,293 person years of follow-up. The cohort start date was December 8, 2020, and the end date was June 30, 2021. The outcome measure we

based on de-identified national clinical records. The English data is stored in the ORCHID TRE. Access to data can be requested via https://orchid.phc.ox. ac.uk/index.php/orchid-data/. The Scottish data is stored in the Public Health Scotland TRE. To access these individual-level, confidential healthcare data, researchers will need to apply to HSC-PBPP (https://www.informationgovernance. scot.nhs.uk/pbpphsc/). The Welsh data are available in the SAIL Databank at Swansea University, Swansea, UK. All proposals to use SAIL data are subject to review by an independent Information Governance Review Panel (IGRP). Before any data can be accessed, approval must be given by the IGRP. The IGRP gives careful consideration to each project to ensure proper and appropriate use of SAIL data. When access has been granted, it is gained through a privacy protecting safe haven and remote access system referred to as the SAIL Gateway. SAIL has established an application process to be followed by anyone who would like to access data via SAIL at https://www.saildatabank.com/application-process.

**Funding:** This research is part of the Data and Connectivity National Core Study, led by Health Data Research UK in partnership with the Office for National Statistics and funded by UK Research and Innovation (grant ref MC_PC_20029, AS). EAVE II is funded by the Medical Research Council (https:// mrc.ukri.org/) (UKRI MC_PC 19075, AS) with the support of BREATHE, The Health Data Research Hub for Respiratory Health (MC_PC_19004, AS), which is funded through the UK Research and Innovation Industrial Strategy Challenge Fund and delivered through Health Data Research UK. This work was supported by the Con-COV team funded by the Medical Research Council (grant number: MR/V028367/1, RL). This work was supported by Health Data Research UK, which receives its funding from HDR UK Ltd (HDR-9006, RL) funded by the UK Medical Research Council, Engineering and Physical Sciences Research Council, Economic and Social Research Council, Department of Health and Social Care (England), Chief Scientist Office of the Scottish Government Health and Social Care Directorates, Health and Social Care Research and Development Division (Welsh Government), Public Health Agency (Northern Ireland), British Heart Foundation (BHF) and the Wellcome Trust. This work was supported by the ADR Wales programme of work (https://www.adruk.org/). The ADR Wales programme of work is aligned to the priority themes as identified in the Welsh Government's national strategy: Prosperity for All. ADR Wales brings together data science experts at Swansea University Medical School, staff from the

examined was incident CVST events recorded in either primary or secondary care records. We carried out a self-controlled case series (SCCS) analysis of this outcome following first dose vaccination with ChAdOx1 and BNT162b2. The observation period consisted of an initial 90-day reference period, followed by a 2-week prerisk period directly prior to vaccination, and a 4-week risk period following vaccination. Counts of CVST cases from each country were tallied, then expanded into a full dataset with 1 row for each individual and observation time period. There was a combined total of 201 incident CVST events in the cohorts (29.5 per million person years). There were 81 CVST events in the observation period among those who a received first dose of ChAdOx1 (approximately 16.34 per million doses) and 40 for those who received a first dose of BNT162b2 (approximately 12.60 per million doses). We fitted conditional Poisson models to estimate incidence rate ratios (IRRs). Vaccination with ChAdOx1 was associated with an elevated risk of incident CVST events in the 28 days following vaccination, IRR = 1.93 (95% confidence interval (CI) 1.20 to 3.11). We did not find an association between BNT162b2 and CVST in the 28 days following vaccination, IRR = 0.78 (95% CI 0.34 to 1.77). Our study had some limitations. The SCCS study design implicitly controls for variables that are constant over the observation period, but also assumes that outcome events are independent of exposure. This assumption may not be satisfied in the case of CVST, firstly because it is a serious adverse event, and secondly because the vaccination programme in the United Kingdom prioritised the clinically extremely vulnerable and those with underlying health conditions, which may have caused a selection effect for individuals more prone to CVST. Although we pooled data from several large datasets, there was still a low number of events, which may have caused imprecision in our estimates.

## Conclusions

In this study, we observed a small elevated risk of CVST events following vaccination with ChAdOx1, but not BNT162b2. Our analysis pooled information from large datasets from England, Scotland, and Wales. This evidence may be useful in risk–benefit analyses of vaccine policies and in providing quantification of risks associated with vaccination to the general public.

---

## Author summary

### Why was this study done?

- There have been indications of a possible association between Coronavirus Disease 2019 (COVID-19) vaccines—in particular, Oxford/AstraZeneca (ChAdOx1)—and cerebral venous sinus thrombosis (CVST), which is a rare type of blood clot in the brain.

- Further evidence on this topic is required in order to inform vaccine policy deliberations.

Wales Institute of Social and Economic Research, Data and Methods (WISERD) at Cardiff University and specialist teams within the Welsh Government to develop new evidence which supports Prosperity for All by using the SAIL Databank at Swansea University, to link and analyse anonymised data. ADR Wales is part of the Economic and Social Research Council (part of UK Research and Innovation) funded ADR UK (grant ES/S007393/1, RL). SVK acknowledges funding from NHS Research Scotland Senior Clinical Fellowship (SCAF/15/02, SVK), the MRC (MC_UU_00022/2, SVK), and the Scottish Government Chief Scientist Office (SPHSU17, SVK). The funders had no role in study design, data collection and analysis, decision to publish, or preparation of the manuscript.

**Competing interests:** I have read the journal's policy and the authors of this manuscript have the following competing interests. AS is a member of the Scottish Government Chief Medical Officer's COVID-19 Advisory Group and the New and Emerging Respiratory Virus Threats (NERVTAG) Risk Stratification Subgroup and AstraZeneca's COVID-19 Thrombocytopenia Taskforce; all roles are remunerated to AS or his institution. AS and SS are members of the editorial board of PLOS Medicine. CRS declares funding from the MRC, NIHR, CSO and New Zealand Ministry for Business, Innovation and Employment and Health Research Council during the conduct of this study. SVK is co-chair of the Scottish Government's Expert Reference Group on COVID-19 and ethnicity, is a member of the Scientific Advisory Group on Emergencies (SAGE) subgroup on ethnicity and acknowledges funding from a NRS Senior Clinical Fellowship, MRC and CSO. CR is a member of the Scottish Government Chief Medical Officer's COVID-19 Advisory Group, SPI-M, MHRA Vaccine Benefit and Risk Working Group. HRS is an advisor to the Scottish Parliament's COVID-19 Committee. RKO is a member of the National Institute for Health and Care Excellence (NICE) Technology Appraisal Committee. DB is employed by Queen's University Belfast, the Public Health Agency and the Department of Health (Northern Ireland). DB is a member of several Northern Ireland and UK government COVID-19 advisory boards, including the Scientific Pandemic Influenza Group on Modelling and the UK Vaccine Effectiveness Expert Panel, and has represented Northern Ireland on the UK Scientific Advisory Group for Emergencies and its subgroups. SdeL through his University holds grants from AstraZeneca, Eli-Lilly, GSK, MSD, Sanofi and Seqirus. He has been advisory board members for Astra Zeneca, Sanofi and Seqirus. MW is a member of UK government COVID-19

## What did the researchers do and find?

- We estimated the rate of CVST events in the 4 weeks following first dose vaccination compared with a 90-day period prior to vaccination.

- The rate of CVST events in the 4 weeks following vaccination with Oxford/AstraZeneca was approximately twice as high compared to the baseline rate, implying an additional 0.25 events on average in this period per million people vaccinated.

- We did not find an association between Pfizer-BioNTech (BNT162b2) and CVST.

## What do these findings mean?

- We found evidence of a slight increased risk of CVST following first dose vaccination with ChAdOx1, but not BNT162b2.

- Public health authorities may wish to take this evidence into account when communicating the benefits and risks associated with COVID-19 vaccines to the general public.

- Our study had some limitations. Although the cohort was large, CVST is an extremely rare event, and this may have caused some imprecision in our estimates. In addition, there is a possibility that our model assumptions were not satisfied.

## Introduction

There have been concerns over possible associations between some Coronavirus Disease 2019 (COVID-19) vaccines and hematological and vascular adverse events, including, in particular, cerebral venous sinus thrombosis (CVST) following ChAdOx1 nCoV-19 (Oxford/AstraZeneca; henceforth ChAdOx1). A number of case reports and case series of CVST following adenovirus vector COVID-19 vaccination have been published [1–5]. A potential link between ChAdOx1 was initially noted by the European Medicines Agency (EMA) in a safety update on March 29, 2021 [6]. On April 8, 2021, the EMA issued an analysis of pharmacovigilance data covering the European Economic Area that found a safety signal for CVST following ChAdOx1 vaccination, risk ratio = 7.73 (95% confidence interval (CI) 5.35 to 10.80) [7]. An observed versus expected analysis using data from the Mayo Clinic Health System in the United States found a relative risk of 1.50 (95% CI 0.28 to 7.10) of CVST in a combined analysis of COVID-19 vaccines [8]. An observed versus expected analysis in Denmark and Norway found a standardised morbidity ratio of 20.25 (95% CI 8.14 to 41.73) for cerebral venous thrombosis following ChAdOx1 vaccination [9]. A self-controlled case series (SCCS) using the QResearch database in England found incidence rate ratio (IRR) in the 8 to 28 days following vaccination of 2.37 (95% CI 1.34 to 4.21) for ChAdOx1 and 1.93 (95% CI 0.87 to 4.28) for BNT162b2 [10]. Several countries suspended ChAdOx1 or restricted its use to older age groups due to these and other safety concerns [11]. The Joint Committee on Vaccination and Immunisation (JCVI), an independent UK-wide body that advises the government on vaccine approval, has recommended that adults under the age of 40 should be offered an alternative to the ChAdOx1 vaccine, if available [12,13].

advisory group, SPI-M, and a member of Scottish
Government COVID-19 Advisory Group. All other
authors report no conflicts of interest.

**Abbreviations:** CHI, Community Health Index; CI,
confidence interval; COVID-19, Coronavirus
Disease 2019; CVST, cerebral venous sinus
thrombosis; EAVE, Early Assessment of Vaccine
and antiviral Effectiveness; EMA, European
Medicines Agency; ICD-10, International
Classification of Diseases-10th Revision; IGRP,
Information Governance Review Panel; IRR,
incidence rate ratio; JCVI, Joint Committee on
Vaccination and Immunisation; RECORD,
REporting of studies Conducted using
Observational Routinely-collected Data; RR, rate
ratio; SCCS, self-controlled case series; TRE,
trusted research environment.

At the time of writing, 3 vaccines are being administered in the UK: ChAdOx1, BNT162b2 (Pfizer-BioNTech), and mRNA-1273 (Moderna). They have all shown high levels of efficacy in Phase II and III clinical trials [14–16]. ChAdOx1 and BNT162b2 have also shown high levels of "real-world effectiveness" against COVID-19 hospitalisation and death [17,18]. Vaccine rollout in the UK started with BNT162b2 on December 8, 2020, followed by ChAdOx1 on January 4, 2021 and mRNA-1273 on April 7, 2021. Guidance from the JCVI included a list of priority groups with the elderly, frontline social and health care workers, and the clinically extremely vulnerable at the highest levels of priority (S2 File). Relatively few people in our cohort were vaccinated with mRNA-1273, so this study focused on BNT162b2 and ChAdOx1 only.

The aim of this study was to investigate possible associations between COVID-19 vaccines and CVST. We carried out a SCCS study of CVST events following first dose vaccination with ChAdOx1 and BNT162b2. The data platform used in this study consisted of linked primary care, secondary care, mortality and virological testing data stored in secure trusted research environments (TREs) in each of England, Scotland, and Wales. CVST is an extremely rare event, with an estimated incidence of 3 to 4 per million person years in adults [19]. In our previous analysis exploring COVID-19 vaccine associations with thrombocytopenic, thromboembolic, and hemorrhagic events using Scottish national data, there were insufficient CVST events to undertake a statistical analysis [20]. We were not able to reliably estimate the association between COVID-19 vaccines and CVST with any individual country-specific dataset due to the low number of events. In order to address this, we pooled incident CVST cases from each of the datasets and carried out a SCCS analysis to estimate IRRs for CVST in those who received a first dose of ChAdOx1 or BNT162b2 vaccines.

## Methods

### Study design and population

We followed a prespecified statistical analysis plan (S3 File). The datasets consisted of linked primary care, secondary care, mortality, and virological testing data stored in secure TREs in each of England, Scotland, and Wales (Fig 1). These data were deterministically linked using unique patient identifiers—NHS number in England and Community Health Index (CHI) number in Scotland. In Wales, a combination of deterministic linkage based on NHS number and probabilistic linkage based personal identifiers was used. Anyone under the age of 16 at the date of event was excluded. A case was defined as anyone in our cohorts with a CVST event following the start of the COVID-19 vaccination programme in the UK. An incident case was defined as the first such event in the cohort time period. The cohort start date was December 8, 2020, and the end date was June 30, 2021.

In accordance with our statistical analysis plan, we initially sought to carry out a meta-analysis of estimates from each country. However, there were too few events in each country for this to be feasible. As a result, we undertook a pooled analyses of aggregate count data from each country. We sought to carry out both a SCCS study and a case–control study using pooled data. However, the case–control study would have required sharing of individual-level information, which was not permitted under the data governance rules implemented by each country's TREs. Therefore, we focused on the pooled SCCS study.

The observation period for the SCCS started 104 days before first dose vaccination and ended 28 days after first dose vaccination (Fig 2). The reference period was defined as the first 90 days of the observation period. The prerisk period was defined as the 14-day period prior to first dose vaccination. The risk period was defined as 0 to 28 days post–first dose vaccination. No individual was censored because we examined incident cases only. We pooled data across the 3 nations on incident cases during the observation period around first dose vaccination

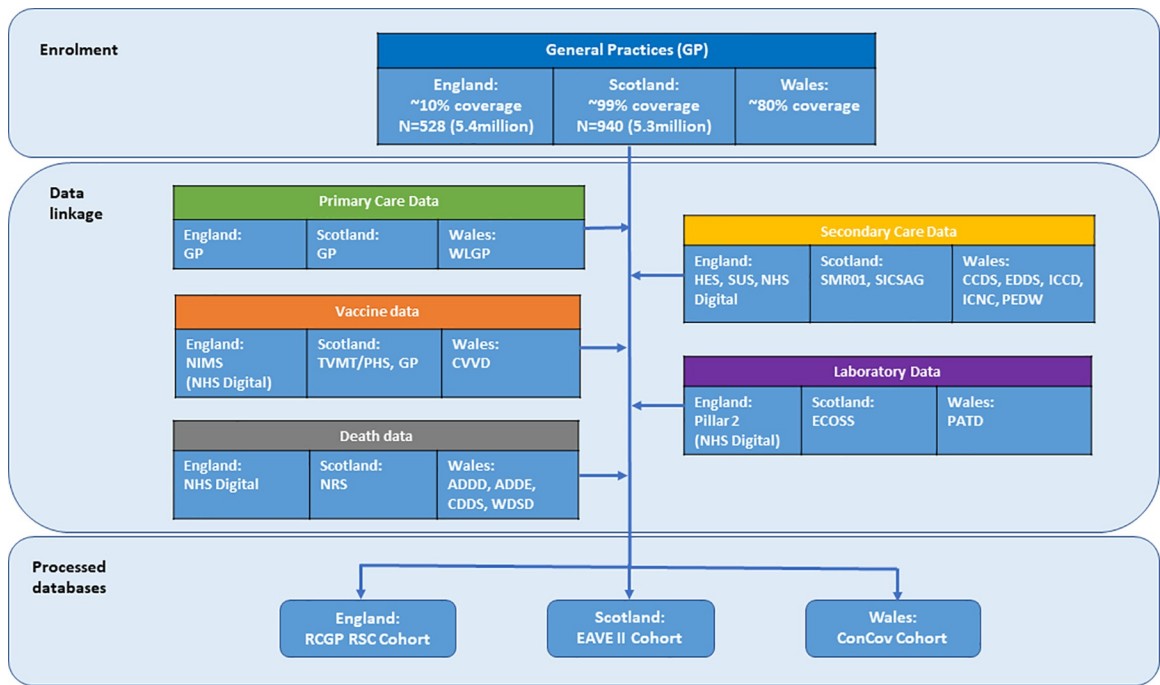

**Fig 1. Data linkage diagram.** ADDD, Annual District Death Daily; ADDE, Annual District Death Extract; CCDS, Critical Care Dataset; CDDS, Consolidated Death Data Source; ConCov, Controlling COVID-19 through enhanced population surveillance and intervention; CVVD, COVID-19 Vaccine Data; EAVE, Early Assessment of Vaccine and antiviral Effectiveness; ECOSS, Electronic Communication of Surveillance in Scotland; EDDS, Emergency Department Dataset; GP, general practices; HES, Hospital Episode Statistics; ICCD, ICNARC – Intensive Care National Audit & Research Centre (COVID only admissions); ICNC, Intensive Care National Audit & Research Centre; NHS, National Health Service; NIMS, National Immunisation Management System; NRS, National Records of Scotland; ONS, Office for National Statistics; PATD, Pathology data COVID-19 Daily; PEDW, Patient Episode Database for Wales; PHS, Publish Health Scotland; RCGP RSC, Oxford-Royal College of General Practitioners Research and Surveillance Centre; SICSAG, Scottish Intensive Care Society Audit Group; SMR01, Scottish Morbidity Records 01; SUS, Secondary Users Service; TVMT, Turas Vaccine Management Tool; WDSD, Welsh Demographic Service Dataset; WLGP, Welsh Longitudinal General Practice Dataset.

with ChAdOx1 or BNT162b2 (Fig 3). Each stratum consisted of an incident case during the reference, prerisk, and risk periods.

## Exposure

We defined an individual as exposed from the day they were administered their first dose of either the BNT162b2 or ChAdOx1 vaccines.

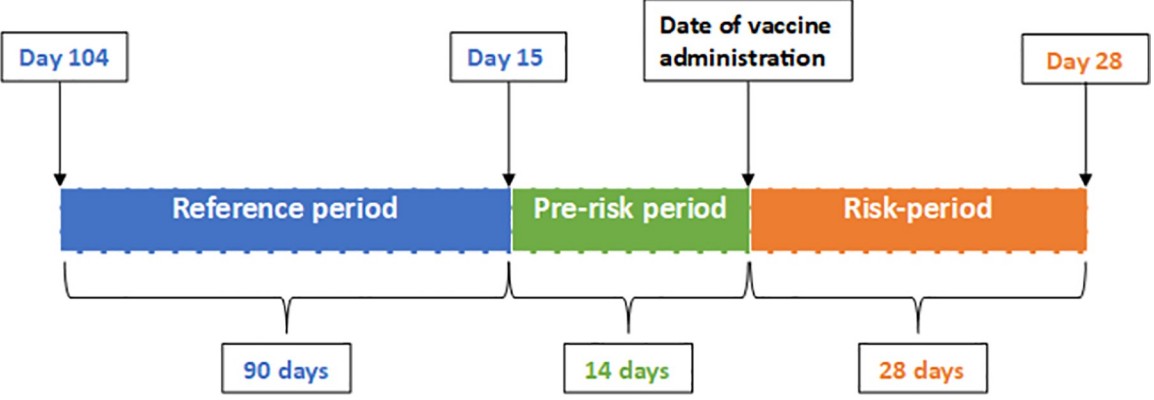

**Fig 2. SCCS study design.** SCCS, self-controlled case series.

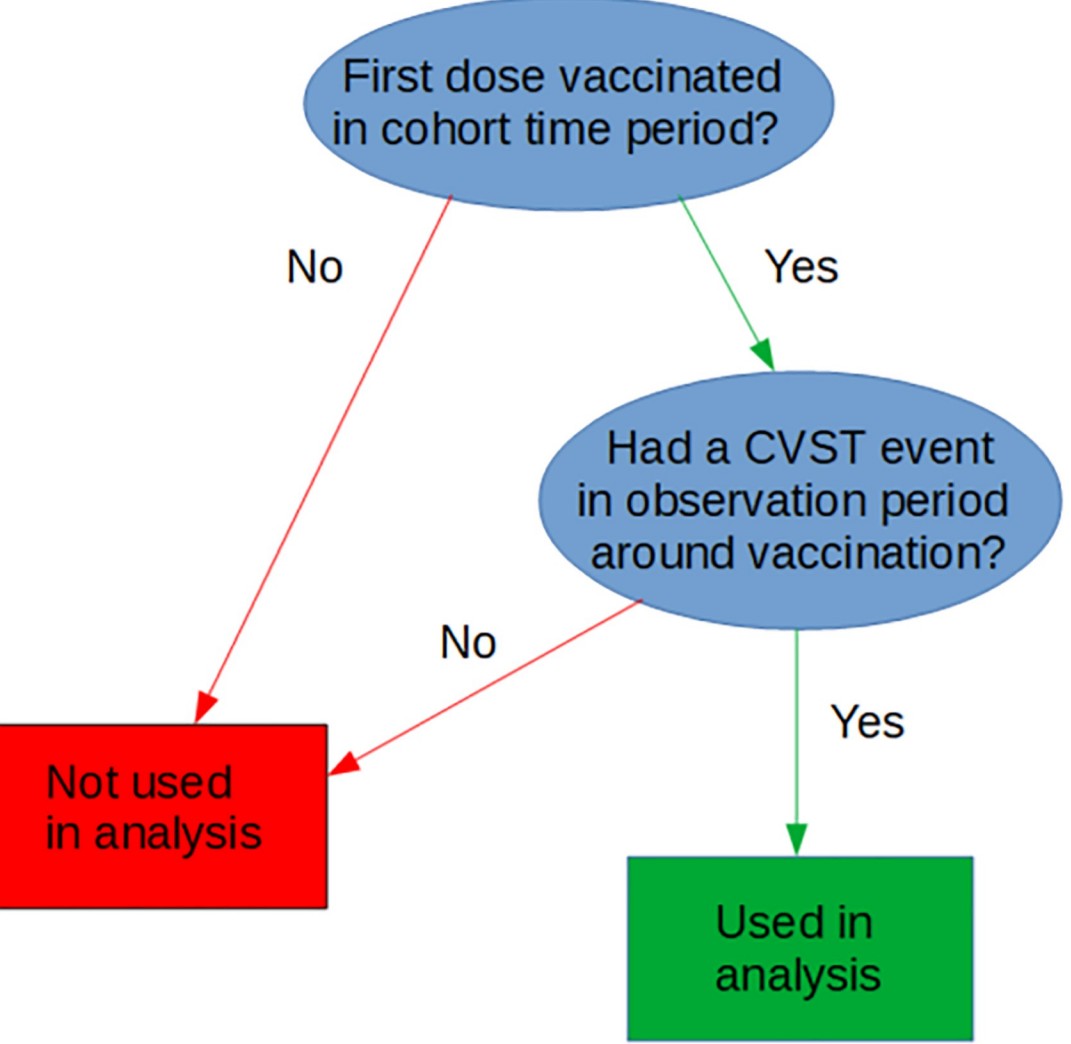

**Fig 3. Selection procedure.** CVST, cerebral venous sinus thrombosis.

## Outcomes

The outcome of interest was incident CVST cases in the observation period. SNOMED codes were used to identify CVST events recorded in primary care electronic health records in England, and Read Codes (Version 2) were used in Scotland and Wales (S4 File). International Classification of Diseases-10th Revision (ICD-10) codes were used to identify CVST events in hospital admission records (S4 File). SNOMED and Read Codes have hierarchies that enable thromboembolic events to be grouped by their location. These enable recording of the diagnosis of an event of interest—for example, a thromboembolic event could be recorded as intracranial, with further subdivisions into CVST, intracranial thrombophlebitis, and thrombosis of cerebral veins, within the SNOMED hierarchy. The code lists used in this study were drawn up by the Early Assessment of Vaccine and antiviral Effectiveness (EAVE) II clinical team and validated by experts in neurology and hematology.

## Statistical analysis

The data governance procedure of the TREs in each country did not allow for individual level data to be shared. However, we obtained permissions to confidentially share counts of

individuals stratified by categories, on the understanding that these would be combined in a pooled count. For a SCCS study of the rate ratios (RRs) of incident events in the observation time periods that does not include any covariates, the only information that is required to estimate the conditional Poisson model is event counts in the reference, prerisk, and risk periods. Analysts in each country had full access to their country's data. Counts of incident cases in the reference, prerisk, and risk periods stratified by vaccine type received were collated in each nation. These were then gathered in the Scottish TRE and expanded into a full dataset with 1 row per individual and observation time period. Synthetic IDs were generated in order to distinguish distinct incident cases in the count data from each other and to label the strata in the model. The process of pooling the data is explained in more detail in the Supporting information (S5 File). IRRs and their 95% CIs were estimated using a conditional Poisson model, with cases during the reference, prerisk, and risk periods as the strata and an offset for the length of each time period. The reference period was taken as the baseline period for calculating IRRs. Conditional Poisson models have a likelihood function that is identical to the corresponding conditional logistic model [21]. Furthermore, incident events can occur at most once. Thus, we used the clogit function for conditional logistic regression from the survival package in R to fit the model. Following a suggestion from one of the reviewers, we undertook 2 post hoc sensitivity analyses: The first excluded any cases who died within 90 days of their event in order to explore the SCCS assumption of event-dependent exposure, and the second focused only on CVST cases identified in secondary care records.

### Ethics and permission

In England, approvals were obtained from the Health Research Authority, London Central (reference number 21/HRA/2786). In Scotland, data approvals were obtained from the National Research Ethics Service Committee, Southeast Scotland 02 (reference number: 12/SS/0201), and Public Benefit and Privacy Panel for Health and Social Care (reference number: 1920–0279). In Wales, approval was provided by SAIL independent Information Governance Review Panel (IGRP) (Project 0911).

### Reporting

This study is reported in accordance with the REporting of studies Conducted using Observational Routinely-collected Data (RECORD) guidelines (S1 File) [22,23].

## Results

The cohorts started on December 8, 2020 and ended on June 30, 2021. Tables 1–3 show the marginal distributions of a number of characteristics in each country's cohort. Clinical risk groups in these tables were derived from the QCovid algorithm [24]. Among the approximately 4.95 million people vaccinated with ChAdOx1, there were 45 incident cases of CVST during the 90-day reference period and 27 incident cases of CVST during the 28-day postvaccine risk period. We found an IRR of 1.93 (95% CI 1.20 to 3.11) in the risk period following first dose vaccination with ChAdOx1 and an IRR of 0.78 (95% CI 0.34 to 1.77) in the risk period following first dose vaccination with BNT162b2 (Table 4). Assuming a baseline incidence of 3 to 4 cases of CVST per million people per year outside of the risk period, our estimates imply an absolute risk of 0.44 to 0.59 incident CVST cases per million people in the 4-week risk period following vaccination with ChAdOx1.

We sought to carry out a post hoc sensitivity analysis focusing on CVST events identified in secondary care records only; however, there were too few events to permit useful estimation of IRRs. We carried out a post hoc sensitivity analysis excluding anyone who died within 90 days

**Table 1. Cohort summary statistics, England.**

| Characteristic | Level | Unvaccinated | One-dose ChAdOx1 | One-dose BNT162b2 |
|---|---|---|---|---|
| Total | | 1,322,305 | 2,247,155 | 1,629,360 |
| Person years of follow-up | | 745,303 | 1,269,229 | 927,224 |
| Incident CVST cases | | 27 | 49 | 30 |
| Deaths | | 16,369 (1.2%) | 11,327 (0.5%) | 8,066 (0.5%) |
| Sex | Female | 649,546 (49.1%) | 1,176,988 (52.4%) | 924,283 (56.7%) |
| | Male | 672,759 (50.9%) | 1,070,167 (47.6%) | 705,077 (43.3%) |
| Age (years) | Mean (SD) | 38.58 (15.54) | 56.49 (14.77) | 52.56 (21.67) |
| Age group (years) | 18 to 64 | 1,222,023 (92.4%) | 1,576,834 (70.2%) | 1,067,807 (65.5%) |
| | 65 to 79 | 69,637 (5.3%) | 558,596 (24.9%) | 318,241 (19.5%) |
| | 80+ | 30,645 (2.3%) | 111,725 (5.0%) | 243,312 (14.9%) |
| Deprivation status[†] | 1 (high) | 303,447 (22.9%) | 323,925 (14.4%) | 232,356 (14.3%) |
| | 2 | 295,143 (22.3%) | 381,384 (17.0%) | 290,696 (17.8%) |
| | 3 | 253,275 (19.2%) | 451,355 (20.1%) | 333,697 (20.5%) |
| | 4 | 239,858 (18.1%) | 512,133 (22.8%) | 368,663 (22.6%) |
| | 5 (low) | 229,464 (17.4%) | 576,815 (25.7%) | 402,905 (24.7%) |
| | Unknown | 1,118 (0.1%) | 1,543 (0.1%) | 1,043 (0.1%) |
| Urban/rural index | Rural town and fringe | 324,138 (14.4%) | 221,355 (13.6%) | 106,384 (8.0%) |
| | Rural town and fringe in a sparse setting | 32,629 (1.5%) | 21,147 (1.3%) | 8,133 (0.6%) |
| | Rural village and dispersed | 51,315 (2.3%) | 34,761 (2.1%) | 19,813 (1.5%) |
| | Rural village and dispersed in a sparse setting | 8,122 (0.4%) | 5,633 (0.3%) | 4,225 (0.3%) |
| | Urban city and town | 1,116,335 (49.7%) | 783,133 (48.1%) | 591,421 (44.7%) |
| | Urban city and town in a sparse setting | 10,593 (0.5%) | 6,858 (0.4%) | 3,384 (0.3%) |
| | Urban major conurbation | 614,524 (27.3%) | 494,346 (30.3%) | 549,592 (41.6%) |
| | Urban minor conurbation | 81,632 (3.6%) | 56,723 (3.5%) | 35,933 (2.7%) |
| | Unknown | 7,867 (0.4%) | 5,404 (0.3%) | 3,420 (0.3%) |
| Number of risk groups[‡] | 0 | 908,219 (68.7%) | 1,165,109 (51.8%) | 807,851 (49.6‡) |
| | 1 | 304,222 (23.0%) | 661,515 (29.4%) | 475,206 (29.2%) |
| | 2 | 76,147 (5.8%) | 255,007 (11.3%) | 199,650 (12.3%) |
| | 3 | 19,985 (1.5%) | 96,980 (4.3%) | 83,191 (5.1%) |
| | 4 | 7,719 (0.6%) | 40,388 (1.8%) | 36,949 (2.3%) |
| | 5+ | 6,013 (0.5%) | 28,156 (1.3%) | 26,513 (1.6%) |
| Number of previous tests[§] | 0 | 1,147,503 (86.8%) | 1,889,412 (84.1%) | 1,350,615 (82.9%) |
| | 1 | 119,612 (9.0%) | 235.388 (10.5%) | 183.933 (11.3%) |
| | 2 | 34,089 (2.6%) | 66,502 (3.0%) | 53,012 (3.3%) |
| | 3 | 10,610 (0.8%) | 23,322 (1.0%) | 17,770 (1.1%) |
| | 4 to 9 | 10,442 (0.8%) | 32,427 (1.4%) | 23,953 (1.5%) |
| | 10+ | 49 (0.0%) | 104 (0.0%) | 77 (0.0%) |
| Average household age | Mean (SD) | 32.6 (17.5) | 47.07 (21.61) | 45.91 (24.88) |
| Number of people in household | 1 | 338,168 (25.6%) | 556,975 (24.8%) | 429,940 (26.4%) |
| | 2 | 261,989 (19.8%) | 698,293 (31.1%) | 499,565 (30.7%) |
| | 3 to 5 | 551,966 (41.7%) | 859,932 (38.3%) | 602,873 (37.0%) |
| | 6 to 10 | 129,832 (9.8%) | 100,041 (4.5%) | 75,591 (4.6%) |
| | 11 to 30 | 18,660 (1.4%) | 17,587 (0.8%) | 9,650 (0.6%) |
| | 31 to 100 | 6,485 (0.5%) | 12,837 (0.6%) | 6,950 (0.4%) |
| | 101+ | 15,205 (1.1%) | 1,490 (0.1%) | 4,791 (0.3%) |

(*Continued*)

**Table 1.** (Continued)

| Characteristic | Level | Unvaccinated | One-dose ChAdOx1 | One-dose BNT162b2 |
|---|---|---|---|---|
| BMI | Underweight | 115,851 (8.8%) | 46,480 (2.1%) | 75,842 (4.7%) |
| | Normal weight | 626,235 (47.4%) | 757,023 (33.7%) | 636,389 (39.1%) |
| | Overweight | 359,299 (27.2%) | 809,120 (36.0%) | 527,780 (32.4%) |
| | Obese | 220,920 (16.7%) | 634,532 (28.2%) | 389,349 (23.9%) |
| Smoking status | Ex-smoker | 308,038 (23.3%) | 657,234 (29.2%) | 438,683 (26.9%) |
| | Nonsmoker | 241,142 (18.2%) | 1,241,281 (55.2%) | 952,839 (58.5%) |
| | Smoker | 714,755 (54.1%) | 338,007 (15.0%) | 209,358 (12.8%) |
| | Unknown | 58,370 (4.4%) | 10,633 (0.5%) | 28,480 (1.7%) |
| Atrial fibrillation | | 11,713 (0.9%) | 68,860 (3.1%) | 74,990 (4.6%) |
| Asthma | | 180,744 (13.7%) | 342,874 (15.3%) | 264,060 (16.2%) |
| Blood cancer | | 4,066 (0.3%) | 20,228 (0.9%) | 18,117 (1.1%) |
| Heart failure | | 8,217 (0.6%) | 40,231 (1.8%) | 39,508 (2.4%) |
| Cerebral palsy | | 724 (0.1%) | 3,326 (0.1%) | 1,442 (0.1%) |
| Coronary heart disease | | 17,102 (1.3%) | 100,989 (4.5%) | 99,116 (6.1%) |
| Cirrhosis | | 1,678 (0.1%) | 6,829 (0.3%) | 4,598 (0.3%) |
| Congenital heart disease | | 4,399 (0.3%) | 15,787 (0.7%) | 10,637 (0.7%) |
| COPD | | 12,266 (0.9%) | 69,891 (3.1%) | 57,883 (3.6%) |
| Dementia | | 7,468 (0.6%) | 29,271 (1.3%) | 23,567 (1.4%) |
| Diabetes type 1 | | 16,158 (0.7%) | 16,158 (0.7%) | 11,773 (0.7%) |
| Diabetes type 2 | | 34,079 (2.6%) | 185,353 (8.2%) | 153,568 (9.4%) |
| Epilepsy | | 22,824 (1.7%) | 52,820 (2.4%) | 33,655 (2.1%) |
| Fracture | | 35,154 (2.7%) | 89,439 (4.0%) | 73,717 (4.5%) |
| Neurological disorder | | 1,964 (0.1%) | 9,885 (0.4%) | 6,276 (0.4%) |
| Parkinson disease | | 1,483 (0.1%) | 8,338 (0.4%) | 6,571 (0.4%) |
| Pulmonary hypertension | | 1,751 (0.1%) | 8,106 (0.4%) | 8,079 (0.5%) |
| Pulmonary rare | | 2,635 (0.2%) | 16,128 (0.7%) | 14,666 (0.9%) |
| Peripheral vascular disease | | 3,856 (0.3%) | 20,518 (0.9%) | 18,554 (1.1%) |
| Rheumatoid arthritis or SLE | | 7,909 (0.6%) | 39,102 (1.7%) | 27,449 (1.7%) |
| Respiratory cancer | | 1,891 (0.1%) | 7,224 (0.3%) | 5,814 (0.4%) |
| Severe mental illness | | 157,557 (11.9%) | 387,974 (17.3%) | 236,108 (14.5%) |
| Sickle cell disease | | 794 (0.1%) | 1,731 (0.1%) | 1,236 (0.1%) |
| Stroke/TIA | | 12,140 (0.9%) | 64,068 (2.9%) | 60,588 (3.7%) |
| Thrombosis or pulmonary embolus | | 5,219 (0.4%) | 23,610 (1.1%) | 18,864 (1.2%) |
| Care housing category | Care home | 5,168 (0.4%) | 21,534 (1.0%) | 10,419 (0.6%) |
| | Homeless | 5,026 (0.4%) | 3,585 (0.2%) | 1,563 (0.1%) |
| Learning disability or Down syndrome | Learning disability | 15,620 (1.2%) | 39,679 (1.8%) | 23,426 (1.4%) |
| | Down syndrome | 271 (0.0%) | 1,857 (0.1%) | 714 (0.0%) |
| Kidney disease | CKD5 without dialysis or transplant | 18,609 (1.4%) | 105,376 (4.7%) | 121,038 (7.4%) |
| | CKD5 with dialysis | 238 (0.0%) | 928 (0.0%) | 812 (0.0%) |
| | CKD5 with transplant | 225 (0.0%) | 1,530 (0.1%) | 1,132 (0.1%) |

[†]Deprivation status: quintiles of UK IMD (2000).

[‡]Number of risk groups: count of QCovid risk groups.

[§]Number of previous tests: proxy for working in a high-risk occupation (e.g., healthcare worker).

BMI, body mass index; CKD, chronic kidney disease; COPD, chronic obstructive pulmonary disease; CVST, cerebral venous sinus thrombosis; IMD, index of multiple deprivation; SD, standard deviation; SLE, systemic lupus erythematosus; TIA, transient ischemic attack.

**Table 2. Cohort summary statistics, Scotland.**

| Characteristic | Levels | Unvaccinated | One-dose ChAdOx1 | One-dose BNT162b2 |
|---|---|---|---|---|
| Total | | 1,837,555 | 1,743,343 | 829,038 |
| Person years of follow-up | | 627,935 | 1,136,652 | 952,284 |
| Incident CVST cases | | 9 | 34 | 13 |
| Deaths | | 14,792 (0.80%) | 12,485 (0.72%) | 6,554 (0.79%) |
| Sex | Female | 847,192 (46.1%) | 900,591 (51.7%) | 519,423 (62.7%) |
| | Male | 990,363 (53.9%) | 842,752 (48.3%) | 309,615 (37.3%) |
| Age (years) | Mean (SD) | 35.8 (13.6) | 59 (15.3) | 57.4 (16.6) |
| Age group (years) | 18 to 64 | 1,756,236 (95.6%) | 1,178,096 (67.6%) | 462,906 (55.8%) |
| | 65 to 79 | 47,113 (2.6%) | 380,175 (21.8%) | 329,739 (39.8%) |
| | 80+ | 34,206 (1.9%) | 185,072 (10.6%) | 36,393 (4.4%) |
| Deprivation status[†] | 1 (high) | 393,735 (21.4%) | 315,337 (18.1%) | 152,365 (18.4%) |
| | 2 | 363,509 (19.8%) | 340,417 (19.5%) | 164,078 (19.8%) |
| | 3 | 345,606 (18.8%) | 361,645 (20.7%) | 166,194 (20.0%) |
| | 4 | 345,959 (18.8%) | 362,057 (20.8%) | 176,278 (21.3%) |
| | 5 (low) | 364,644 (19.8%) | 355,356 (20.4%) | 165,303 (19.9%) |
| | Unknown | 24,101 (1.3%) | 8,531 (0.5%) | 4,820 (0.6%) |
| Urban/rural index | 1—large urban areas | 779,728 (42.4%) | 527,068 (30.2%) | 246,393 (29.7%) |
| | 2—other urban areas | 592,533 (32.2%) | 652,571 (37.4%) | 334,968 (40.4%) |
| | 3—accessible small towns | 148,961 (8.1%) | 178,437 (10.2%) | 80,831 (9.7%) |
| | 4—remote small towns | 72,218 (3.9%) | 98,686 (5.7%) | 42,336 (5.1%) |
| | 5—accessible rural | 147,110 (8.0%) | 179,672 (10.3%) | 71,206 (8.6%) |
| | 6—remote rural | 72,905 (4.0%) | 98,378 (5.6%) | 48,484 (5.8%) |
| | Unknown | 24,101 (1.3%) | 8,531 (0.5%) | 4,820 (0.6%) |
| Number of risk group[‡] | 0 | 1,380,273 (75.1%) | 812,869 (46.6%) | 411,082 (49.6%) |
| | 1 | 370,157 (20.1%) | 519,923 (29.8%) | 241,381 (29.1%) |
| | 2 | 65,651 (3.6%) | 238,303 (13.7%) | 103,522 (12.5%) |
| | 3 | 13,232 (0.7%) | 100,913 (5.8%) | 42,747 (5.2%) |
| | 4 | 4,740 (0.3%) | 43,232 (2.5%) | 18,217 (2.2%) |
| | 5+ | 3,502 (0.2%) | 28,103 (1.6%) | 12,089 (1.5%) |
| Number of previous tests[§] | 0 | 1,521,965 (82.8%) | 1,450,847 (83.2%) | 599,480 (72.3%) |
| | 1 | 237,779 (12.9%) | 204,180 (11.7%) | 114,003 (13.8%) |
| | 2 | 46,071 (2.5%) | 46,784 (2.7%) | 33,299 (4.0%) |
| | 3 | 11,350 (0.6%) | 15,396 (0.9%) | 14,173 (1.7%) |
| | 4 to 9 | 11,842 (0.6%) | 18,816 (1.1%) | 27,847 (3.4%) |
| | 10+ | 8,548 (0.5%) | 7,320 (0.4%) | 40,236 (4.9%) |
| Average household age | Mean (SD) | 34.4 (13.8) | 54.8 (17.8) | 53.7 (18.5) |
| Number of people in household[¶] | 1 | 545,823 (29.7%) | 613,519 (35.2%) | 259,370 (31.3%) |
| | 2 | 404,714 (22.0%) | 603,417 (34.6%) | 298,158 (36.0%) |
| | 3 to 5 | 784,498 (42.7%) | 493,935 (28.3%) | 236,711 (28.6%) |
| | 6 to 10 | 82,513 (4.5%) | 30,138 (1.7%) | 15,493 (1.9%) |
| | 11 to 30 | 8,569 (0.5%) | 1,532 (0.1%) | 6,669 (0.8%) |
| | 31 to 100 | 6,255 (0.3%) | 600 (0.0%) | 10,967 (1.3%) |
| | 101+ | 5,182 (0.3%) | 202 (0.0%) | 1,670 (0.2%) |
| BMI | Underweight | 23,875 (1.3%) | 17,726 (1.0%) | 7,908 (1.0%) |
| | Normal weight | 234,944 (12.8%) | 214,888 (12.3%) | 108,885 (13.1%) |
| | Overweight | 1,422,006 (77.4%) | 1,108,478 (63.6%) | 519,249 (62.6%) |
| | Obese | 156,730 (8.5%) | 402,251 (23.1%) | 192,996 (23.3%) |

(*Continued*)

**Table 2.** (Continued)

| Characteristic | Levels | Unvaccinated | One-dose ChAdOx1 | One-dose BNT162b2 |
|---|---|---|---|---|
| Smoking status | Ex-smoker | 143,236 (7.8%) | 297,503 (17.1%) | 138,702 (16.7%) |
| | Nonsmoker | 700,880 (38.1%) | 669,699 (38.4%) | 328,683 (39.6%) |
| | Smoker | 299,528 (16.3%) | 430,035 (24.7%) | 192,753 (23.3%) |
| | Unknown | 693,910 (37.8%) | 346,106 (19.9%) | 168,900 (20.4%) |
| Atrial fibrillation | | 6,353 (0.3%) | 73,050 (4.2%) | 29,466 (3.6%) |
| Asthma | | 224,636 (12.2%) | 243,888 (14.0%) | 109,010 (13.1%) |
| Blood cancer | | 1,772 (0.1%) | 14,751 (0.8%) | 5,541 (0.7%) |
| Heart failure | | 3,448 (0.2%) | 33,261 (1.9%) | 12,293 (1.5%) |
| Cerebral palsy | | 664 (0.0%) | 4,334 (0.2%) | 1,072 (0.1%) |
| Coronary heart disease | | 11,859 (0.6%) | 133,205 (7.6%) | 60,514 (7.3%) |
| Cirrhosis | | 3,483 (0.2%) | 14,263 (0.8%) | 6,204 (0.7%) |
| Congenital heart disease | | 3,586 (0.2%) | 24,379 (1.4%) | 10,277 (1.2%) |
| COPD | | 9,670 (0.5%) | 90,514 (5.2%) | 35,037 (4.2%) |
| Dementia | | 3,276 (0.2%) | 17,779 (1.0%) | 17,286 (2.1%) |
| Diabetes type 1 | | 2,005 (0.1%) | 14,889 (0.9%) | 5,589 (0.7%) |
| Diabetes type 2 | | 16,641 (0.9%) | 167,766 (9.6%) | 79,735 (9.6%) |
| Epilepsy | | 9,160 (0.5%) | 43,185 (2.5%) | 12,989 (1.6%) |
| Fracture | | 55,777 (3.0%) | 97,098 (5.6%) | 43,284 (5.2%) |
| Neurological disorder | | 1,433 (0.1%) | 12,562 (0.7%) | 4,561 (0.6%) |
| Parkinson disease | | 626 (0.0%) | 6,061 (0.3%) | 3,093 (0.4%) |
| Pulmonary hypertension | | 709 (0.0%) | 6,523 (0.4%) | 1,722 (0.2%) |
| Pulmonary rare | | 1,399 (0.1%) | 16,060 (0.9%) | 5,849 (0.7%) |
| Peripheral vascular disease | | 3,441 (0.2%) | 28,820 (1.7%) | 12,450 (1.5%) |
| Rheumatoid arthritis or SLE | | 3,414 (0.2%) | 31,267 (1.8%) | 12,947 (1.6%) |
| Respiratory cancer | | 1,218 (0.1%) | 7,243 (0.4%) | 2,552 (0.3%) |
| Severe mental illness | | 161,104 (8.8%) | 263,637 (15.1%) | 119,114 (14.4%) |
| Sickle cell disease | | 388 (0.0%) | 1,905 (0.1%) | 762 (0.1%) |
| Stroke/TIA | | 7,931 (0.4%) | 80,030 (4.6%) | 35,481 (4.3%) |
| Thrombosis or pulmonary embolus | | 8,086 (0.4%) | 50,760 (2.9%) | 18,625 (2.2%) |
| Care housing category | Care home | 2,277 (0.1%) | 2,806 (0.2%) | 16,583 (2.0%) |
| | Homeless | 2,271 (0.1%) | 1,360 (0.1%) | 342 (0.0%) |
| Learning disability or Down syndrome | Learning disability | 24,695 (1.3%) | 34,031 (2.0%) | 9,551 (1.2%) |
| | Down syndrome | 116 (0.0%) | 1,193 (0.1%) | 391 (0.0%) |
| Kidney disease | CKD5 without dialysis or transplant | 7,558 (0.4%) | 103,725 (5.9%) | 41,568 (5.0%) |
| | CKD5 with dialysis | 546 (0.0%) | 4,335 (0.2%) | 1,394 (0.2%) |
| | CKD5 with transplant | 370 (0.0%) | 3,107 (0.2%) | 1,279 (0.2%) |

[†]Deprivation status: quintiles of SIMD 2020.

[‡]Number of risk groups: count of QCovid risk groups.

[§]Number of previous tests: proxy for working in a high-risk occupation (e.g., healthcare worker).

[¶]Household information taken from September 2020.

BMI, body mass index; CKD, chronic kidney disease; COPD, chronic obstructive pulmonary disease; CVST, cerebral venous sinus thrombosis; SD, standard deviation; SIMD, Scottish Index of Multiple Deprivation; SLE, systemic lupus erythematosus; TIA, transient ischemic attack.

of their CVST event. The only estimate that changed from our main analysis was that for the risk period following vaccination with ChAdOx1, the IRR was 1.79 (95% CI 1.10 to 2.91) (S6 File).

**Table 3. Cohort summary statistics, Wales.**

| Characteristic | Levels | Unvaccinated | One-dose ChAdOx1 | One-dose BNT162b2 |
|---|---|---|---|---|
| Total | | 346,683 | 964,494 | 717,224 |
| Person years of follow-up | | 205,886 | 541,057 | 402,723 |
| Incident CVST cases | | 6 | 24 | 9 |
| Deaths | | 7,741 (2.2%) | 6,876 (0.7%) | 1,388 (0.2%) |
| Sex | Male | 197,387 (56.9%) | 476,300 (49.4%) | 329,736 (46.0%) |
| | Female | 149,296 (43.1%) | 488,194 (50.6%) | 387,488 (54.0%) |
| Age (years) | Mean (SD) | 36.27 (17.10) | 58.49 (15.94) | 44.82 (19.60) |
| Age group (years) | 18 to 64 | 274,386 (91.3%) | 636,502 (66.0%) | 538,559 (75.9%) |
| | 65 to 79 | 17,389 (5.8%) | 209,653 (21.7%) | 158,915 (22.4%) |
| | 80+ | 8,739 (2.9%) | 117,968 (12.2%) | 12,278 (1.7%) |
| Deprivation status[†] | 1 (most deprived) | 94,765 (27.3%) | 183,424 (19.0%) | 143,271 (20.0%) |
| | 2 | 74,747 (21.6%) | 195,260 (20.2%) | 146,938 (20.5%) |
| | 3 | 69,401 (20.0%) | 190,358 (19.7%) | 131,847 (18.4%) |
| | 4 | 53,523 (15.4%) | 191,504 (19.9%) | 131,130 (18.3%) |
| | 5 (least deprived) | 54,247 (15.6%) | 203,948 (21.1%) | 164,038 (22.9%) |
| Number of risk groups[‡] | 1 | 129,012 (37.2%) | 181,916 (18.9%) | 211,942 (29.6%) |
| | 2 | 122,262 (35.3%) | 305,529 (31.7%) | 256,138 (35.7%) |
| | 3 | 60,691 (17.5%) | 239,308 (24.8%) | 150,040 (20.9%) |
| | 4 | 22,745 (6.6%) | 129,906 (13.5%) | 61,613 (8.6%) |
| | 5+ | 11,973 (3.5%) | 107,835 (11.2%) | 37,491 (5.2%) |
| Number of previous tests[§] | 0 | 289,816 (83.6%) | 787,529 (81.7%) | 525,062 (73.2%) |
| | 1 | 39,404 (11.4%) | 122,007 (12.6%) | 123,853 (17.3%) |
| | 2 | 9,074 (2.6%) | 28,177 (2.9%) | 32,648 (4.6%) |
| | 3 | 2,676 (0.8%) | 9,090 (0.9%) | 9,544 (1.3%) |
| | 4 to 9 | 3,371 (1.0%) | 11,425 (1.2%) | 10,037 (1.4%) |
| Average household age | Mean (SD) | 36.12 (15.09) | 52.28 (18.83) | 42.83 (18.64) |
| Number of people in household | 1 | 35,087 (10.1%) | 164,573 (17.1%) | 77,093 (10.8%) |
| | 2 | 59,280 (17.1%) | 322,717 (33.5%) | 190,182 (26.5%) |
| | 3 to 5 | 191,646 (55.4%) | 413,605 (42.9%) | 389,868 (54.4%) |
| | 6 to 10 | 52,028 (15.0%) | 47,698 (4.9%) | 53,992 (7.5%) |
| | 11 to 30 | 5,817 (1.7%) | 10,467 (1.1%) | 4,431 (0.6%) |
| | 31 to 100 | 1,164 (0.3%) | 4,356 (0.5%) | 882 (0.1%) |
| | 101+ | 830 (0.2%) | 183 (0.0%) | 408 (0.1%) |
| BMI | Normal weight | 137,044 (39.8%) | 238,075 (24.7%) | 214,268 (29.9%) |
| | Obese | 75,615 (21.9%) | 374,711 (38.9%) | 237,832 (33.2%) |
| | Overweight | 103,442 (30.0%) | 325,633 (33.8%) | 237,648 (33.2%) |
| | Underweight | 28,634 (8.3%) | 25,261 (2.6%) | 26,418 (3.7%) |
| Smoking status | Ex-smoker | 37,314 (10.8%) | 231,039 (24.0%) | 127,107 (17.7%) |
| | Nonsmoker | 140,892 (40.6%) | 512,280 (53.1%) | 402,210 (56.1%) |
| | Smoker | 98,437 (28.4%) | 201,499 (20.9%) | 136,068 (19.0%) |
| | Unknown | 70,040 (20.2%) | 19,676 (2.0%) | 51,839 (7.2%) |
| Atrial fibrillation | | 3,005 (0.9%) | 43,976 (4.6%) | 15,274 (2.1%) |
| Asthma | | 45,591 (13.2%) | 147,572 (15.3%) | 110,270 (15.4%) |
| Blood cancer | | 699 (0.2%) | 7,116 (0.7%) | 2,886 (0.4%) |
| Heart failure | | 1,816 (0.5%) | 22,248 (2.3%) | 6,879 (1.0%) |
| Cirrhosis | | 586 (0.2%) | 4,903 (0.5%) | 1,889 (0.3%) |
| Congestive heart disease | | 4,482 (1.3%) | 61,338 (6.4%) | 22,283 (3.1%) |

(*Continued*)

**Table 3.** (Continued)

| Characteristic | Levels | Unvaccinated | One-dose ChAdOx1 | One-dose BNT162b2 |
|---|---|---|---|---|
| COPD | | 3,794 (1.1%) | 44,984 (4.7%) | 16,158 (2.3%) |
| Dementia | | 1,438 (0.4%) | 13,155 (1.4%) | 2,540 (0.4%) |
| Diabetes type I | | 714 (0.2%) | 6,034 (0.6%) | 1,628 (0.2%) |
| Diabetes type II | | 8,081 (2.3%) | 108,610 (11.3%) | 36,764 (5.1%) |
| Epilepsy | | 3,295 (1.0%) | 18,776 (1.9%) | 5,136 (0.7%) |
| Fracture | | 11,205 (3.2%) | 40,808 (4.2%) | 26,066 (3.6%) |
| Neurological disorder | | 439 (0.1%) | 4,260 (0.4%) | 1,154 (0.2%) |
| Parkinson disease | | 313 (0.1%) | 3,845 (0.4%) | 1,312 (0.2%) |
| Pulmonary hypertension | | 254 (0.1%) | 2,550 (0.3%) | 904 (0.1%) |
| Pulmonary rare | | 528 (0.2%) | 6,287 (0.7%) | 2,755 (0.4%) |
| Peripheral vascular disease | | 1,107 (0.3%) | 12,368 (1.3%) | 4,547 (0.6%) |
| Rheumatoid arthritis | | 1,376 (0.4%) | 14,754 (1.5%) | 6,454 (0.9%) |
| Respiratory cancer | | 495 (0.1%) | 3,620 (0.4%) | 1,621 (0.2%) |
| Severe mental illness | | 42,890 (12.4%) | 153,083 (15.9%) | 87,343 (12.2%) |
| Stroke | | 3,175 (0.9%) | 39,032 (4.0%) | 12,706 (1.8%) |
| Thrombosis or pulmonary embolus | | 3,468 (1.0%) | 30,726 (3.2%) | 10,296 (1.4%) |
| Care housing category | Care home | 1,150 (0.3%) | 7,729 (0.8%) | 1,375 (0.2%) |
| | Homeless | 2,308 (0.7%) | 2,431 (0.3%) | 1,230 (0.2%) |
| Learning disability or Down syndrome | Learning disability | 7,232 (2.1%) | 16,250 (1.7%) | 9,397 (1.3%) |
| | Down syndrome | 25 (0.0%) | 195 (0.0%) | 74 (0.0%) |
| Kidney disease | CKD5 without dialysis or transplant | 277 (0.1%) | 2,711 (0.3%) | 802 (0.1%) |
| | CKD5 with transplant | 69 (0.0%) | 921 (0.1%) | 364 (0.1%) |
| | CKD5 with dialysis | 47 (0.0%) | 304 (0.0%) | 61 (0.0%) |

[†]Deprivation status: WIMD 2020.

[‡]Number of risk groups: Individual QCovid.

[§]Number of previous tests: Proxy for working in a high-risk occupation (e.g., healthcare worker).

BMI, body mass index; CKD, chronic kidney disease; COPD, chronic obstructive pulmonary disease; CVST, cerebral venous sinus thrombosis; SD, standard deviation; SLE, systemic lupus erythematosus; TIA, transient ischemic attack; WIMD, Welsh index of multiple deprivation.

**Table 4. Number of events and IRRs for CVST following first dose vaccination with ChAdOx1 and BNT162b2.**

| Time period | Number of events | IRR (95% CI) |
|---|---|---|
| ChAdOx1 | | |
| Reference | 45 | 1 |
| Prerisk | 9 | 1.29 (0.63 to 2.63) |
| Risk | 27 | 1.93 (1.20 to 3.11) |
| BNT162b2 | | |
| Reference | 29 | 1 |
| Prerisk | <5 | 0.89 (0.31 to 2.52) |
| Risk | 7 | 0.78 (0.34 to 1.77) |

Event counts of <5 have been suppressed in accordance with disclosure control principles implemented by the data controllers. With the day of vaccination as day 0, the reference period was day −104 to day −14. The prerisk period was day −14 to day 0. The risk period was day 0 to day 28.

CI, confidence interval; CVST, cerebral venous sinus thrombosis; IRR, incidence rate ratio.

Within each nation, we plotted histograms of event counts in the observation period in order to explore the suitability of the prerisk period to account for event-dependent exposure to vaccination. There appeared to be a reduction in events in the 2-week prerisk period, but no obvious reduction outside of that time period. We are unable to share these histograms due to the statistical disclosure principles enacted by the TREs that hosted this analysis.

## Discussion

Our pooled SCCS analysis of national datasets from England, Scotland, and Wales found an elevated risk of CVST in the 4-week period following vaccination with ChAdOx1. We did not find an association between BNT162b2 and CVST.

There have hitherto been few population-based studies estimating the association between COVID-19 vaccines and CVST. Such analyses are made challenging by the extreme rarity of CVST. Our study corroborates previous observed–expected analyses from the EMA [7], the Mayo Clinic Health System [8], a population-based study in Denmark and Norway [9], and a SCCS using the QResearch database in England [10]. Our results are broadly consistent with the latter study [10], as would be expected due to the similar study design and population. Our estimated RRs are, however, notably smaller than risk ratios reported in [7–9]. This is likely due to the fact that we conducted a SCCS that implicitly controls for variables that are constant over the observation period, as opposed to an observed–expected analysis.

A limitation in the SCCS analysis is the assumption that occurrence of an event does not affect subsequent exposure [25]. CVST is a serious adverse event that can be life threatening. Therefore, the assumption of event-independent exposure may not have been satisfied. This could have caused a selection effect where individuals who were more likely to have a CVST event were less likely to be vaccinated and thus less likely to be included in our analysis. A large multinational study of prognosis for CVST found a death rate of 4.3% and a death or dependency rate of 18.9% at hospital discharge, where dependency was defined as a score of >2 on the modified Rankin scale [26]. On the other hand, among the JCVI priority groups for vaccination are "clinically extremely vulnerable individuals" (group 4), and "all individuals aged 16 to 64 years with underlying health conditions which put them at higher risk of serious disease and mortality" (group 6) (S2 File). It is possible that the vaccination programme created a selection effect for vaccination of people more prone to CVST events.

In order to explore the validity of the assumption of event-independent exposure, we carried out a sensitivity analysis excluding those who died within 90 days of their event. This had a small effect on the IRR for the risk period following vaccination with ChAdOx1, with all other estimates unchanged. We also included a prerisk period in the SCCS study design to account for the possibility that occurrence of a CVST event affected subsequent vaccination and plotted histograms of event counts over time in order to assess its suitability. Our statistical analysis plan included a case–control analysis, which may have been useful to further explore this assumption. However, this would have required sharing more detailed data between TREs, which was not possible under the permissions in place.

We estimated IRRs as opposed to RRs. Estimating RRs with pooled data across the 3 nations could be achieved if we were able to share individual-level data. However, this was not permitted under the statistical disclosure rules implemented by the data controllers. We do not believe that the estimates for RRs and IRRs would be significantly different because of the rarity of CVST events.

We plan to extend our analysis to mRNA-1273 and to second and booster dose vaccinations. To date, there have been very few studies that have estimated the association between COVID-19 vaccine and CVST events using large-scale nationally representative datasets [7–9].

Although we had access to a large, combined cohort, there were still relatively few events. Further evidence corroborating our results is required.

In conclusion, we found an increased risk of CVST following first dose vaccination with ChAdOx1. We did not find an increased risk following first dose vaccination with BNT162b2. This evidence may be useful in risk–benefit evaluations for vaccine-related policies and in providing quantification of risks associated with vaccination to the general public.

## Supporting information

**S1 File. STROBE and RECORD checklists.** RECORD, REporting of studies Conducted using Observational Routinely-collected Data; STROBE, STrengthening the Reporting of OBservational studies in Epidemiology.
(DOCX)

**S2 File. Vaccine priority groups.** Context of vaccine roll-out in the UK: JCVI COVID-19 vaccination priority group list. COVID-19, Coronavirus Disease 2019; JCVI, Joint Committee on Vaccination and Immunisation.
(DOCX)

**S3 File. Statistical analysis plan.**
(DOCX)

**S4 File. Code lists.** Read Codes and SNOMED CT codes for CVST. CVST, cerebral venous sinus thrombosis.
(DOCX)

**S5 File. Data pooling procedure.**
(DOCX)

**S6 File. Sensitivity analysis.**
(DOCX)

## Acknowledgments

We thank Dave Kelly from Albasoft for his support with making primary care data available and James Pickett, Wendy Inglis-Humphrey, Vicky Hammersley, Maria Georgiou, and Laura Gonzalez Rienda for their support with project management and administration. Our thanks to J. Quint, R. Al-Shahi Salman, and Q. Hill for reviewing code lists. This work uses data provided by patients and collected by the NHS as part of their care and support. We would also like to acknowledge all data providers who make anonymised data available for research. We wish to acknowledge the collaborative partnership that enabled acquisition and access to the deidentified data in Wales, which led to this output. The collaboration was led by the Swansea University Health Data Research UK team under the direction of the Welsh Government Technical Advisory Cell and includes the following groups and organisations: the Secure Anonymised Information Linkage (SAIL) Databank, Administrative Data Research Wales, NHS Wales Informatics Service, Public Health Wales, NHS Shared Services Partnership, and the Welsh Ambulance Service Trust. All research conducted has been completed under the permission and approval of the SAIL independent IGRP (project number: 0911).

## Author Contributions

**Conceptualization:** Aziz Sheikh.

**Data curation:** Steven Kerr, Mark Joy, Fatemeh Torabi, Stuart Bedston, Utkarsh Agrawal, Declan Bradley, Emily Lowthian, James Marple, Dylan McGagh, Emily Moore, Jiafeng Pan, Syed Ahmar Shah, Ting Shi, Ruby S. M. Tsang, Eleftheria Vasileiou, Chris Robertson.

**Formal analysis:** Steven Kerr, Mark Joy, Fatemeh Torabi, Stuart Bedston, Utkarsh Agrawal, Declan Bradley, Emily Moore, Jiafeng Pan, Eleftheria Vasileiou, Chris Robertson.

**Investigation:** Antony Chuter.

**Methodology:** Steven Kerr, Chris Robertson.

**Project administration:** Ashley Akbari, Jillian Beggs, Annemarie B. Docherty, David Ford, Richard Hobbs, Simon de Lusignan, Ronan Lyons, Jim McMenamin, Josephine-L. K. Murray, Mark Woolhouse, Colin R. Simpson, Aziz Sheikh.

**Supervision:** Steven Kerr, Annemarie B. Docherty, David Ford, Srinivasa Vittal Katikireddi, Colin McCowan, Rhiannon K. Owen, Lewis Ritchie, Sarah Stock, Chris Robertson.

**Writing – original draft:** Steven Kerr.

**Writing – review & editing:** Steven Kerr, Ashley Akbari, Jillian Beggs, Declan Bradley, Antony Chuter, Annemarie B. Docherty, David Ford, Richard Hobbs, Srinivasa Vittal Katikireddi, Emily Lowthian, Simon de Lusignan, Ronan Lyons, James Marple, Colin McCowan, Dylan McGagh, Jim McMenamin, Emily Moore, Josephine-L. K. Murray, Rhiannon K. Owen, Jiafeng Pan, Lewis Ritchie, Syed Ahmar Shah, Ting Shi, Sarah Stock, Ruby S. M. Tsang, Eleftheria Vasileiou, Mark Woolhouse, Colin R. Simpson, Chris Robertson, Aziz Sheikh.

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
