## [Editor Report · Decision Letter 0]

23 Aug 2021

Dear Dr Kerr, 

Thank you for submitting your manuscript entitled "First dose ChAdOx1 and BNT162b2 COVID-19 vaccinations and cerebral venous sinus thrombosis: pooled self-controlled case series of UK datasets" for consideration by PLOS Medicine.

Your manuscript has now been evaluated by the PLOS Medicine editorial staff and I am writing to let you know that we would like to send your submission out for external peer review.

Kind regards,

Louise Gaynor-Brook, MBBS PhD

Senior Editor

PLOS Medicine

---

## [Decision Letter · Decision Letter 1]

22 Sep 2021

Dear Dr. Kerr,

Thank you very much for submitting your manuscript "First dose ChAdOx1 and BNT162b2 COVID-19 vaccinations and cerebral venous sinus thrombosis: pooled self-controlled case series of UK datasets" (PMEDICINE-D-21-03598R1) for consideration at PLOS Medicine. 

Your paper was evaluated by three independent reviewers, including a statistical reviewer, and was discussed among all the editors here and with an academic editor with relevant expertise. The reviews are appended at the bottom of this email and any accompanying reviewer attachments can be seen via the link below:

[LINK]

In light of these reviews, I am afraid that we will not be able to accept the manuscript for publication in the journal in its current form, but we would like to consider a revised version that addresses the reviewers' and editors' comments. Obviously we cannot make any decision about publication until we have seen the revised manuscript and your response, and we plan to seek re-review by one or more of the reviewers. 

We expect to receive your revised manuscript by Oct 13 2021 11:59PM. Please email us (plosmedicine@plos.org) if you have any questions or concerns.

We look forward to receiving your revised manuscript. 

Sincerely,

Louise Gaynor-Brook, MBBS PhD

Associate Editor 

PLOS Medicine

plosmedicine.org

General comments:

Throughout the paper, please adapt reference call-outs to the following style: "... ChAdOx1 vaccine, if available [11,12]." (noting the absence of spaces within the square brackets).

Title: Please revise your title according to PLOS Medicine's style. We suggest “First dose ChAdOx1 and BNT162b2 COVID-19 vaccinations and cerebral venous sinus thrombosis: A pooled self-controlled case series study of 12 million individuals in England, Scotland and Wales” or similar

Abstract:

Abstract Background: Please specify which COVID-19 vaccines were restricted. “England, Scotland and Wales” would be preferable to “Great Britain” 

Abstract Methods and Findings:

Please provide brief demographic details of the study population (e.g. sex, age, ethnicity, etc)

Please provide the exact number of participants, population, the dates between which the study took place, length of follow up, and main outcome measures.

If possible, please provide the actual incidence of CVST in the study population in addition to IRRs 

Line 62 - Please revise to “Vaccination with ChAdOx1 was associated with an elevated risk…” 

In the last sentence of the Abstract Methods and Findings section, please describe 2-3 of the main limitations of the study's methodology.

Abstract Conclusions:

Please begin your Abstract Conclusions with "In this study, we observed ..." or similar, to summarize the main findings from your study, without overstating your conclusions. Please emphasize what is new and address the specific implications of your study, being careful to avoid assertions of primacy and general statements such as "these results have implications for policy". 

Author Summary:

In the final bullet point of ‘What Do These Findings Mean?’, please describe the main limitations of the study in non-technical language.

Methods:

Please state that your study had a prospective statistical analysis plan early in the Methods section, and please include the relevant prospectively written document with your revised manuscript as a Supporting Information file to be published alongside your study, and cite it in the Methods section. A legend for this file should be included at the end of your manuscript. Changes in the analysis-- including those made in response to peer review comments-- should be identified as such in the Methods section of the paper, with rationale. If a reported analysis was performed based on an interesting but unanticipated pattern in the data, please be clear that the analysis was data-driven.

Please add the following statement, or similar, to the Methods: "This study is reported as per the REporting of studies Conducted using Observational Routinely-collected Data (RECORD) guideline (S1 Checklist)." 

Line 153 - sentence appears incomplete 

Results: 

Please provide a table showing the baseline characteristics of the study population as Table 1. 

Please replace ~ with approximately

Line 184 - please revise to ‘incident cases of CVST’

Please report the dates between which the study took place

Discussion:

Please present and organize the Discussion as follows: a short, clear summary of the article's findings; what the study adds to existing research and where and why the results may differ from previous research; strengths and limitations of the study; implications and next steps for research, clinical practice, and/or public policy; one-paragraph conclusion.

Line 202 - please temper assertions of primacy by adding ‘to the best of our knowledge’ or similar. “England, Scotland and Wales” would be preferable to “Great Britain” 

Tables:

Please define all abbreviations used in the table legend of each table.

References:

Please ensure that journal name abbreviations match those found in the National Center for Biotechnology Information (NCBI) databases, and are appropriately formatted and capitalised.

Please also see https://journals.plos.org/plosmedicine/s/submission-guidelines#loc-references for further details on reference formatting, e.g. six authors are names prior to ‘et al’. 

Where websites are cited, please specify the date of access 

Supplementary files: 

Please see https://journals.plos.org/plosmedicine/s/supporting-information for our supporting information guidelines. 

Comments from the reviewers:

Reviewer #1: "First dose ChAdOx1 and BNT162b2 COVID-19 vaccinations and cerebral venous sinus thrombosis: pooled self-controlled case series of UK datasets" is a concise self-controlled case series (SCCS) analysis on incident cerebral venous sinus thrombosis (CVST) events, on over 12 million individuals from England, Scotland and Wales. The extremely low incidences for CVST (<50 for 4.33 million ChAdOx1 doses, <15 for 2.32 million BNT162b2 doses) has made any conclusive determination of increased risk challenging. With the SCCS design that implicitly controls for time-invariant variables (by effectively comparing every individual against himself before [including a clearance period] and after dosing), a 2.66 (1.48-4.79) incidence rate ratio (IRR) was found for ChAdOx1, against 1.07 (0.22-5.31) for BNT162b2 (Table 2).

While the study framework appears to allow for the analysis of other potential side effects in addition to CVST, this submission remains a timely addition to the literature, with further work on mRNA-1273 and further doses already proposed (Line 236). Still, a few issues might be considered:

1. While the description of the data compilation is relatively comprehensive (Figure 1), the detailed selection of the cohort might stand to be defined in greater detail. For example, the starting point (all individual given at least one dose of either the ChAdOx1 or BNT162b2 vaccine?) and any exclusions (e.g. missing data) might be presented in a flowchart.

2. While simple cohort demographics are provided in Table 1, it appears relevant to also present other demographic variables (e.g. age, gender, ethnicity, etc.) stratified by vaccine received, which would moreover help to give an idea of whether the demographics for each vaccine are comparable (all the more as their administration appears non-random from S1 Table).

Moreover, it appears possible that the underlying demographics may remain important in the analysis of IRR (related to the selection bias raised in Line 218), despite the SCCS study design. For example, assume that a rare disease has effectively zero incidence below a certain age (e.g. 50 years). Then, if Treatment A were provided mostly to those below that age, and Treatment B mostly to those above that age, then the potential for incidence would appear biased towards Treatment B (before considerations on whether increased disease incidence due to vaccine dosing is itself affected by age and other variables). Notably, from Table 2, given the 25 pre-risk events for ChAdOx1 (4.33 million doses), about 13 pre-risk events would be expected for BNT162b2 (2.32 million doses). However, only 6 pre-risk events were observed for BNT162b2 in reality. A fuller evaluation of the demographics for each vaccine group might therefore be warranted.

3. Data coverage would seem to remain somewhat of a concern, especially for the English data (~10% GP coverage, from Figure 1). In particular, the validity of SCCS would seem to assume individuals remaining "in system" throughout, i.e. if CVST were to occur for an individual, this CVST incidence would be expected to be recorded whether in the pre-risk, clearance or risk period. Would this be a reasonable expectation, or is it possible that CVST be treated at a venue that is not included in the databases used (e.g. the ~90% of GP practices not covered, for England)?

Reviewer #2: Thank you for the opportunity to review this manuscript on a matter of importance to public health.

Major comments: 

1. 34 authors? Please elaborate.

2. Overall, the manuscript appears rushed and needs more attention. See e.g. line 153?

3. I do not understand the statistical approach. It is unclear what aggregation in each country and then expansion with so-called synthetic ID's is. If this means that you have first aggregated the true events and then tried to reconstruct synthetic events I am concerned. Please provide more detail and references that this is a valid approach. Alternatively, please estimate the association in each country and combine the three using e.g. inverse variance weighting.

4. Please provide sensitivity analyses supporting that the assumptions of the SCCS analysis is not violated.

5. Please provide a figure with two panels: each panel showing a histogram of the timing of event compared to date of vaccination with the two vaccines.

6. You need to convince the reader that the many SNOMED and READ codes you have included accurately reflect CVST. Have these been validated? 

7. Are you including GP information? Can you please provide a sensitivity analysis of only cases identified in a hospital setting.

8. What are you using laboratory data for?

9. You need to discuss your results in more detail in comparison to other studies. You report a <1 CVST case per 1 mil vaccinated. This is much less than what others have reported.

10. You conclude "Public health officials should take account of this evidence in designing policies on vaccination." What do you mean? Elaborate please?

Minor comments:

11. incidence rate ratio instead of incident rate ratio 

12. I am confused by the Sup table 1 mentioning Scotland only? 

13. In SCCS terminology, a pre-risk period is your clearance period and the reference period is your pre-risk period.

Reviewer #3: This is a straightforward analysis and methodological sound. However, I fail to see the novelty of the data as the association between CVT and vaccination has already been reported. Nonetheless, the study provides estimates for the British population. 

Some important data is missing and could add relevant information. Please provide data on mortality, demographic differences, or association with coagulation disorders. 

You might consider presenting you findings as a brief report or as a letter.

The figure legends are somewhat misplaced in the text.

[LINK]

---

## [Decision Letter · Decision Letter 2]

1 Dec 2021

Dear Dr. Kerr,

Thank you very much for submitting your manuscript "First dose ChAdOx1 and BNT162b2 COVID-19 vaccinations and cerebral venous sinus thrombosis: A pooled self-controlled case series study of 11.6 million individuals in England, Scotland and Wales" (PMEDICINE-D-21-03598R2) for consideration at PLOS Medicine. 

Your paper was re-reviewed by three reviewers and discussed among all the editors here. The reviews are appended at the bottom of this email and any accompanying reviewer attachments can be seen via the link below:

[LINK]

In light of the statistical re-review, I am afraid that we will not be able to accept the manuscript for publication in the journal in its current form, but we would like to consider a revised version that addresses the reviewers' and editors' comments. Obviously we cannot make any decision about publication until we have seen the revised manuscript and your response, and we plan to seek re-review by one or more of the reviewers. 

We expect to receive your revised manuscript by Dec 22 2021 11:59PM. Please email us (plosmedicine@plos.org) if you have any questions or concerns.

We look forward to receiving your revised manuscript. 

Sincerely,

Louise Gaynor-Brook, MBBS PhD

Associate Editor, PLOS Medicine

plosmedicine.org

It is of concern that the results of your analysis appear to have changed substantially during your revision (Table 4) - in particular, the differences in numbers of events. Please provide clarification as to how the revised results were generated, including changes in the data used and methodology. This may be subject to statistical re-review. 

Comments from the reviewers:

Reviewer #1: Figure 2 defining the pre-risk, clearance and risk periods appears unchanged from the previous revision (R1). However Table 4 for reference/pre-risk/risk events/incidence rate ratios appears wholly different from Table 2 from the previous revision (R1). This might be clarified, since it appears the core of the analysis.

Reviewer #2: The authors have satisfactorily addressed my comments. Thank you.

Reviewer #3: all well addressed

[LINK]

---

## [Editor Report · Decision Letter 3]

14 Jan 2022

Dear Dr. Kerr,

Thank you very much for re-submitting your manuscript "First dose ChAdOx1 and BNT162b2 COVID-19 vaccinations and cerebral venous sinus thrombosis: A pooled self-controlled case series study of 11.6 million individuals in England, Scotland and Wales" (PMEDICINE-D-21-03598R3) for review by PLOS Medicine.

I have discussed the paper with my colleagues and the academic editor. I am pleased to say that provided the remaining editorial and production issues are dealt with we are planning to accept the paper for publication in the journal.

The remaining issues that need to be addressed are listed at the end of this email. Please take these into account before resubmitting your manuscript:

[LINK]

In revising the manuscript for further consideration here, please ensure you address the specific points made by the editors. In your rebuttal letter you should indicate your response to the editors' comments and the changes you have made in the manuscript. Please submit a clean version of the paper as the main article file. A version with changes marked must also be uploaded as a marked up manuscript file. Please also check the guidelines for revised papers at http://journals.plos.org/plosmedicine/s/revising-your-manuscript for any that apply to your paper. 

We look forward to receiving the revised manuscript by Jan 21 2022 11:59PM.   

Sincerely,

Louise Gaynor-Brook, MBBS PhD

Associate Editor 

PLOS Medicine

plosmedicine.org

Requests from Editors:

Abstract Methods and Findings:

Please revise to “6,808,293 person years of follow-up”

If possible, please provide a brief demographic summary of the study population(s) (e.g. % female, mean age (SD), etc)

Please specifically use the term ‘limitations’ in section beginning “The self-controlled case series study design…”

Author Summary:

Please revise to “following first dose vaccination with either ChAdOx1 or Pfizer BioNTech (BNT162b2)”, or similar 

Please separate the text under ‘What Do These Findings Mean?’ into two separate bullet points. 

Please add a final bullet point to ‘What Do These Findings Mean?’ to describe the main limitations of the study in non-technical language.

Introduction:

Please revise to “...United States found a relative risk of 1.50 (95% CI 0.28-7.10) of CVST …”

Methods

Please provide your STROBE/RECORD checklist as (S1 Checklist)

Results: 

Please revise “4.95 million people total vaccinated”

PLOS does not permit ‘data not shown’; please provide the results generated in your sensitivity analysis (perhaps in a supplementary file) 

Tables:

Please define all abbreviations used in the table legend of each table e.g. COPD, BMI, CVST and so on.

Table 1 - please clarify what is meant by ‘Age (years%)’; please clarify whether mean(SD) and Median(IQR) are presented and whether % signs are appropriate for Age and Average Household Age 

Table 2 - please clarify what is meant by ‘Age (years%)’

Table 4 - please provide a reminder in the table legend as to what the different time periods represent (i.e. reference, pre-risk, risk) 

References:

Please try to be consistent in reference formatting - a DOI number for the full-text article is acceptable as an alternative to or in addition to traditional volume and page numbers. 

Please remove italicised text e.g. ref 11 and report journal names as found at http://www.ncbi.nlm.nih.gov/nlmcatalog/journals e.g. Lancet (ref 15), N Engl J Med (refs 1, 2, etc) 

Please see https://journals.plos.org/plosmedicine/s/submission-guidelines#loc-references for further details on reference formatting. 

Supplementary files: 

Please provide your STROBE/RECORD checklist as (S1 Checklist)

[LINK]

---

## [Editor Report · Decision Letter 4]

21 Jan 2022

Dear Dr Kerr, 

On behalf of my colleagues and the Academic Editor, Prof. Suzanne Cannegieter, I am pleased to inform you that we have agreed to publish your manuscript "First dose ChAdOx1 and BNT162b2 COVID-19 vaccinations and cerebral venous sinus thrombosis: A pooled self-controlled case series study of 11.6 million individuals in England, Scotland and Wales" (PMEDICINE-D-21-03598R4) in PLOS Medicine.

PRESS

Sincerely, 

Louise Gaynor-Brook, MBBS PhD 

Associate Editor 

PLOS Medicine